# Whole-brain mapping of socially isolated zebrafish reveals that lonely fish are not loners

**Hande Tunbak[1], Mireya Vazquez-Prada[1], Thomas Michael Ryan[1], Adam Raymond Kampff[2], Elena Dreosti[1]***

[1]The Wolfson Institute for Biomedical Research, University Street, University College London, London, United Kingdom; [2]Sainsbury Wellcome Centre, Howland Street, University College London, London, United Kingdom

**Abstract** The zebrafish was used to assess the impact of social isolation on behaviour and brain function. As in humans and other social species, early social deprivation reduced social preference in juvenile zebrafish. Whole-brain functional maps of anti-social isolated (lonely) fish were distinct from anti-social (loner) fish found in the normal population. These isolation-induced activity changes revealed profound disruption of neural activity in brain areas linked to social behaviour, social cue processing, and anxiety/stress. Several of the affected regions are modulated by serotonin, and we found that social preference in isolated fish could be rescued by acutely reducing serotonin levels.

**\*For correspondence:**
e.dreosti@ucl.ac.uk

**Competing interests:** The authors declare that no competing interests exist.

## Introduction

Social preference behaviour, the drive for individuals to identify and approach members of their own species (*Rogers-Carter et al., 2018*; *Winslow, 2003*), is a fundamental component of all social behaviour. We previously found that most zebrafish develop a strong social preference by 2–3 weeks of age (*Dreosti et al., 2015*), yet we also found a small number (~10%) of 'loner' fish that were averse to social cues. A similar diversity of individual social preferences has been found in many species, including humans (*Sloan Wilson et al., 1994*). Loneliness, undesired isolation from social interaction, has been linked to a reduction in social preference (*Engeszer et al., 2004*; *Shams et al., 2018*). We therefore asked whether the socially-averse *loner* fish found in the normal population would show a similar behavioural phenotype and neuronal activity to socially-averse *lonely* fish raised in isolation. To answer this question, we compared the behavioural and functional responses of isolated fish to controls during viewing of conspecifics. This comparison found that isolation induces patterns of brain activity that are not present in the normal population. We then asked if we could rescue the aversive behaviour of isolated fish. Since some of the highly activated areas in isolated fish are serotoninergic, we used Buspirone, a 5HT1A receptor agonist. These findings will have important implications for how we understand and treat the impact of social isolation.

Prolonged periods of social isolation are particularly detrimental to humans during early development. However, even brief periods of social isolation have been shown to impact mental and physical health. We therefore tested two models of social isolation, Full (fish raised completely without social interaction) and Partial (fish isolated for 48 hr prior to behavioural testing). Each experiment comprised two sessions, 15 min of acclimation to the chamber followed by 15 min of exposure to two size matched sibling fish that were not isolated. To quantify social preference, we calculated a visual preference index (VPI) that compares the amount of time fish spend in the chamber nearest the conspecifics versus the opposite chamber where they are visually isolated from social cues (see Materials and methods). Full social isolation (Fi) caused a significant decrease in social preference relative to normally raised sibling controls (C) (*Figure 1A*, left and middle panel: C vs Fi, p=8.3e$^{-8}$,

**eLife digest** Socialising is good for people's mental health and wellbeing. The connections and relationships that we form can make us more resilient and healthier. Researchers also know that prolonged periods of social isolation, and feeling lonely, can be detrimental to our health, especially in early childhood. The paradox is that loneliness often results in an even lower desire for social contact, leading to further isolation. But not everyone craves social contact. Some people prefer to be alone and feel more comfortable avoiding social interaction.

Zebrafish display the same social preferences. This, along with their transparent brains, makes them a useful model to study the links between social behaviour and brain activity. Like humans, zebrafish are social animals, with most fish taking a strong liking to social interactions by the time they are a few weeks old. A small number of 'loner' fish, however, prefer to avoid interacting with their siblings or tank mates. And so, if loneliness quells the desire for more social contact, the question becomes, does isolation turn otherwise social fish into loners?

Here, Tunbak et al. use zebrafish to study how social isolation changes brain activity and behaviour. Social fish were isolated from others in the tank for a few days. These so-called 'lonely fish' were then allowed back in contact with the other fish. This revealed that, after isolation, previously social fish did avoid interacting with others.

With this experimental set-up, Tunbak et al. also compared the brains of lonely and loner fish. When fish that prefer social interaction were deprived of social contact, they had increased activity in areas of the brain related to stress and anxiety. These lonely fish became anxious and very sensitive to stimuli; and their brain activity suggested that social interaction became overwhelming rather than rewarding. Positively, the lonely fish quickly recovered their normal, social behaviour when given a drug that reduces anxiety.

This work provides a glimpse into how human behaviour could be affected by lengthy periods in isolation. These results suggest that humans could feel anxious upon returning to normal life after spending a long time alone. Moreover, the findings show the impact that social interaction and isolation can have on the young, developing brain.

Mann-Whitney). Specifically, there was an increase in the number of individuals that had a large negative VPI. We therefore decided to divide the fish into three sociality groups: a) anti-social (-S) fish with VPIs below $-0.5$; b) pro-social (+S) fish with VPIs above $+0.5$; c) non-social fish with $-0.5 < VPI < +0.5$. Fish that underwent Partial isolation (Pi), exhibited an intermediate, yet highly significant, change in social preference (*Figure 1A*, right panel: C vs Pi, p=2.5e$^{-8}$, Mann-Whitney).

As previously reported (*Zellner et al., 2011*), we found that fish raised in isolation were significantly less active than their normally raised siblings during the acclimation period (*Figure 1B*: C vs Fi, p=9.0e$^{-6}$; C vs Pi, p=2.8e-$^9$ Mann-Whitney) and during the social viewing session (*Figure 1—figure supplement 1A*: left C vs Fi, p=0.0001; C vs Pi, p=0.004 Mann-Whitney). We then divided fish into groups based on their social preference. Interestingly, anti-social fully and partially isolated fish showed very similar movement activity compared to anti-social controls during the acclimation (*Figure 1C* left: C (-S) vs Fi (-S), p=0.17 Mann-Whitney; C (-S) vs Pi (-S) p=0.23 Mann-Whitney) and during the social viewing session (*Figure 1—figure supplement 1B* left: C (-S) vs Fi (-S), p=0.48 Mann-Whitney; C (-S) vs Pi (-S) p=0.10 Mann-Whitney). The pro-social isolated fish, which also exhibited a reduction in activity relative to controls during the acclimation session (*Figure 1C* right: C (-S) vs Fi (-S), p=8.0e$^{-5}$ Mann-Whitney; C (-S) vs Pi (-S) p=1.0e$^{-7}$ Mann-Whitney), instead showed similar activity relative to controls during social viewing (*Figure 1—figure supplement 1B* right: C (-S) vs Fi (-S), p=0.02 Mann-Whitney; C (-S) vs Pi (-S) p=0.14 Mann-Whitney). In addition, we noticed that all isolated fish behaved qualitatively differently, exhibiting prolonged periods of quiescence (freezing) even when observing conspecifics (*Figure 1D* and *Video 1*).

Freezing is a hallmark of anxiety-like behaviour observed in many species, and reported in zebrafish exposed to stressors (*Giacomini et al., 2015*; *Shams et al., 2018*), including periods of social isolation (*Egan et al., 2009*; *Shams et al., 2017*). In order to quantify freezing behaviour, we measured the percentage of time spent in continuous periods (>3 s) without motion (*Figure 1—figure supplement 1C-D*). We found that both fully and partially isolated fish exhibited significantly more

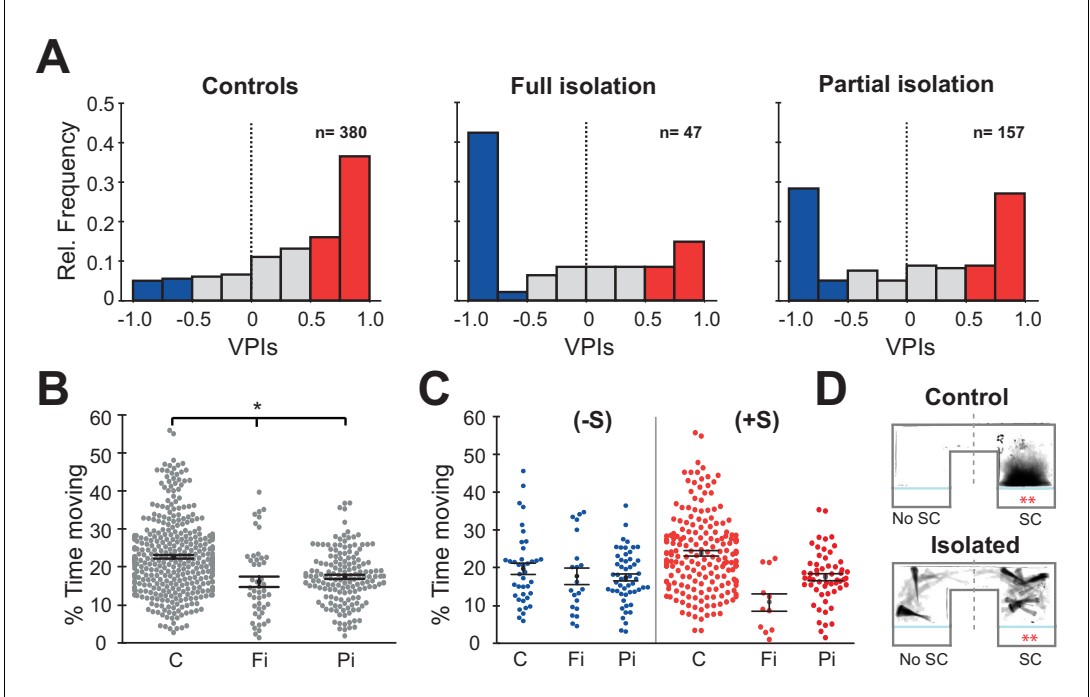

**Figure 1.** Isolation alters social preference behavior and swimming activity. (**A**) Histograms of all the VPIs during the social cue period across different conditions: controls (C, left), full isolation (Fi, middle), and partial isolation (Pi, right). For visual clarity, red bars highlight strong pro-social fish (+S, VPIs > 0.5), blue bars anti-social fish(-S, VPIs < -0.5), and gray non-social fish (ns, -0.5 < VPI < +0.5). (**B**) Swarm plots comparing the activity levels of fish during the acclimation period expressed as percent time moving (C, n=380; Fi, n=47; Pi, n=157). Mean and standard errors are shown. (**C**) Swarm plots comparing the activity levels of anti-social (left) and social (fish) fish during visual social cue exposure for each rearing condition (C (-S), n=39; Fi (-S), n=21; Pi (-S), n=53) or (C (+S), n=193; Fi (+S), n=11; Pi (+S), n=57). (**D**) Time projection through the video of a pro-social control, C(+S), and a fully isolated, Fi (+S), fish during social cue exposure. The dashed lines mark the division between the social cue side (SC) and the side without social cues (No SC) that was used to calculate VPI.

The online version of this article includes the following figure supplement(s) for figure 1:

**Figure supplement 1.** Isolation alters social preference behaviour and swimming activity.

freezing than controls during the acclimation period (*Figure 1—figure supplement 1C* left: C vs Fi, p=$3.4e^{-16}$ Mann-Whitney; C vs Pi, p=$2.8e^{-5}$ Mann-Whitney), and that this increase relative to controls persisted for fully isolated fish during social viewing, but was reduced in partially isolated fish, perhaps representing some recovery during the 15 min of social interaction (*Figure 1—figure supplement 1D* left: C vs Fi, p=$6.3e^{-13}$ Mann-Whitney; C vs Pi, p=0.03 Mann-Whitney). When we compared freezing behaviour of groups with similar social preference, we found, as expected, that anti-social fish exhibited increased freezing during social viewing regardless of rearing condition. However, pro-social fully isolated fish also showed increased freezing during social viewing, suggesting that they were not engaged in typical social interaction, but rather remained immobile on the side with the conspecifics (*Figure 1—figure supplement 1D* right).

The behavioural similarities between anti-social isolated (*lonely*) and anti-social control (*loner*) fish led us to hypothesize that isolation might simply predispose fish to the same anti-social state found in the normal population. If this is the case, neural activity of anti-social isolated and anti-social control fish should be similar when presented with social cues. To test this hypothesis, we performed whole-brain two-photon imaging of *c-fos* expression, an immediate early gene whose expression is associated with increased neural activity (*Herrera and Robertson, 1996*), in juvenile brains following testing in the social preference assay. Dissected brains were imaged with the dorsal surface down (bottom-up) to achieve clear views of the ventral brain structures that have been previously implicated in the social brain network (*Figure 2A*, also see Materials and methods). Volumes of 1.5 mm x 1.5 mm x 700 µm, with a voxel size of 1 × 1×3 µm, were acquired from 135 zebrafish brains across all experimental groups and registered to a reference brain (*Marquart et al., 2017*). These *c-fos*

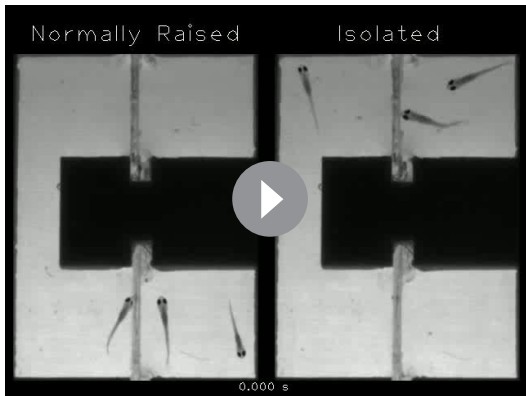

**Video 1.** Example of a control and a fully isolated +S fish video during social cue presentation. Two minutes of behaviour is shown in 20 s (6x playback acceleration). The control fish shows a strong social preference for the social cue and has a stereotypical social phenotype (left). The test fish spends most of its time watching the social cue with a 45-degree angle and synchronizing its bout motion with the other two conspecifics. The fully isolated fish spends long periods of time as well on the side of the conspecifics. Its behaviour, however, its characterized by long pauses while watching the conspecifics (right).
https://elifesciences.org/articles/55863#video1

whole-brain functional maps were first normalised to a background intensity level (see Materials and methods) and then used to compare the neural activity patterns of different test groups. We compared the average activity map for each rearing/sociality condition with the average map acquired from similarly raised sibling fish that were placed in the behavioural assay for 30 min without any social cues (nsc, no social-cue). The resulting normalised difference stacks (e.g. (+S - nsc)/nsc) allowed us to identify changes in neural activity associated with exposure to a visual social cue (*Figure 2A*).

Several brain areas showed strong activation or inhibition in normally raised fish upon social cue exposure. We focused on areas that have been reported as social brain areas (*O'Connell and Hofmann, 2011*) and show differences between our experimental groups (*Figure 2B*: C (+S and -S)). The caudal hypothalamus was differentially activated in pro- vs. anti-social control fish. A dorsal sub-region was significantly activated in pro-social controls (*Figure 2B and D*: dHc - C (+S) vs C (nsc), p=0.007, Mann-Whitney), whereas it was inhibited, along with the adjacent ventral sub-region, in anti-social controls (*Figure 2B and D*: vHc - C (-S) vs C (nsc), p=0.003, Mann-Whitney). The caudal hypothalamus is known to express high levels of serotonin and dopamine, as well as glutamate and histamine (*Filippi et al., 2010*; *Kaslin and Panula, 2001*). Furthermore, a segregation into distinct dorsal and ventral areas of the caudal hypothalamus has already been shown for some of these markers, such as tyrosine hydroxylase 1 and 2, (Th1 and Th2) (*Yamamoto et al., 2010*) and we confirmed these previous results with immunostaining (*Figure 2C* left), as well as for the dopamine and serotonin transporters, DAT and slc6a4b (*Figure 2C* right) (*Filippi et al., 2010*; *Lillesaar, 2011*). Changes in serotonin and dopamine levels have been widely documented in response to social interaction (*Scerbina et al., 2012*), viewing social cues (*Saif et al., 2013*), and social isolation (*Huang et al., 2015*; *Shams et al., 2018*; *Shams et al., 2015*). While the serotonergic system has been linked to stress and arousal (*Backström and Winberg, 2017*), the dopamine circuitry has been shown to regulate the reward system underlying social behaviour (*Teles et al., 2013*). Since the caudal hypothalamus expresses both of these neurotransmitters, and our data demonstrate a pattern of activation/inhibition that is distinct for pro- and anti-social fish, then this area could be crucial in regulating social preference.

The second social brain area we investigated was the preoptic area. Our data showed a similar activation pattern for anti-social and pro-social fish characterised by a small increase in the dorsal preoptic area (dPa) and a small decrease in the ventral preoptic area (vPa). However, only anti-social control fish showed a significant change in the ventral area (*Figure 2B and D*: C (-S) vs C (nsc), vPa p=0.003, Mann-Whitney). The activation of the preoptic area during social behaviour is consistent with previous literature in a number of species (*O'Connell and Hofmann, 2011*). This area has been shown to express several neuropeptides involved in social behaviour such as arginine/vasotocin and oxytocin (*Heinrichs et al., 2009*; *Herget and Ryu, 2015*). It was recently shown that oxytocin does not seem to be responsible for social interaction (*Ribeiro et al., 2019*) as mutants for oxytocin receptors shows no alteration in social preference, but rather reduced social recognition. Furthermore, injections of oxytocin do not have any effect on shoaling and interaction (*Langen et al., 2015*). The neuropeptide vasotocin, instead, has been shown to have a specific effect on reducing social interaction (*Langen et al., 2015*) and not shoaling behaviour. This neuropeptide has also been shown to be involved in aggression (*Teles et al., 2016*) and stress by stimulating cortisol release.

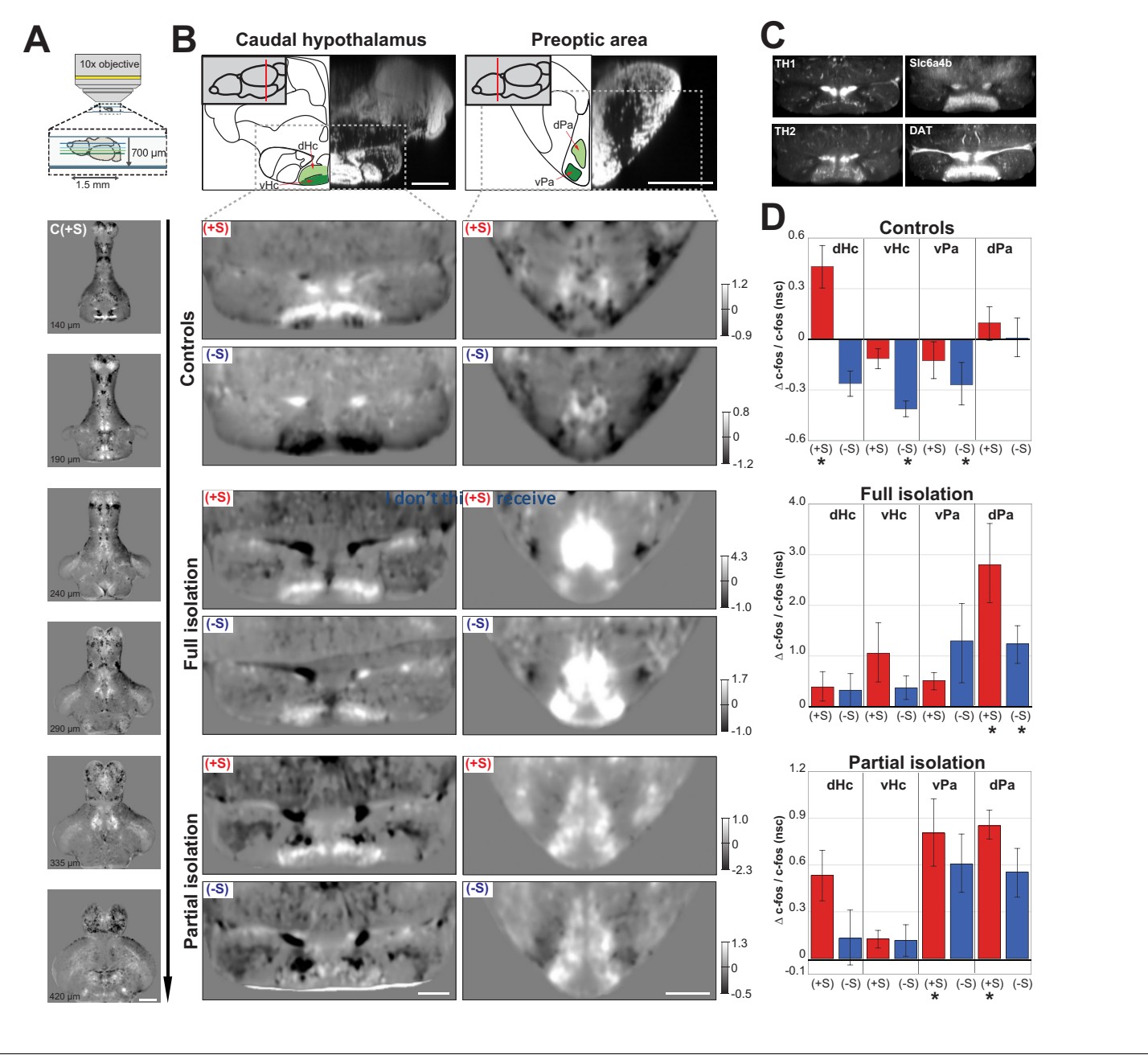

**Figure 2.** Functional maps of the social brain in normal and isolated fish. (A) Schematic of the custom-built two-photon microscope used for acquiring whole-brain volumes of dorsal-down mounted fish brains (top panel). Horizontal sections of pro-social control fish (C(+S)) responses at increasing imaging depth (lower panels). Images are average differences between (C(+S)) and siblings not presented with a social cue. Positive values (white) indicate increased cFos expression in socially preferring fish, while negative values (black) indicate decreased expression. Scale bar is 200μm. The intensity scale bar is shown in B, C(+S) row. (B) Region analysis of two different brain areas that have been implicated in social behavior: caudal hypothalamus and preoptic area. A schematic of the anatomical regions and corresponding DAPI staining is shown (top panel) with two sub-regions highlighted in green. Images showing changes in cFos activation in these areas for pro- (+S) and anti-social (-S) controls, fully isolated, and partially isolated fish are shown. Images are horizontal sections of the average difference between each test group and their corresponding sibling group not presented with a social cue. Scale bar is 100μm. Intensity scale bar is shown for each group. (C) Average image of TH1, TH2, Slc6a4b, and DAT expression in the same section of the caudal hypothalamus as 2B (n=3 each). Scale bar is 100μm. (D) Summary graphs showing the change in cFos activation for four different brain areas calculated by using the average difference images shown in (B) and using 3D masks (a single plane of each area of the masks is shown in green in B). Positive values indicate increases in cFos expression; asterisks mark significant changes relative to no social cue siblings. D=dorsal and V=ventral caudal hypothalamus; Pa=ventrolateral preoptic area, PM=dorsal preoptic area.

We then compared the brain activity maps of anti- and pro-social control fish with fully and partially isolated fish. As described previously, anti-social control (*loner*) fish showed a behavioural phenotype very similar to anti-social isolated (*lonely*) fish. Therefore, we investigated whether their brain activity maps were also similar following the presentation of a social cue. Contrary to our hypothesis, *c-fos* functional maps of anti-social fully isolated fish (*Figure 2B*: Fi (-S)) revealed a completely different activity profile than their anti-social sibling controls (*Figure 2B*: C (-S)). The ventral sub-region of the caudal hypothalamus (vHc) of Fi (-S) fish was not inactivated, while the preoptic area was strongly activated in both the dorsal (dPa) and the ventral (vPa) regions, but significantly only in the dorsal (*Figure 2B and D*: Fi (-S) vs Fi (nsc), p=0.006 dPa; p=0.07 vPa, Mann-Whitney). Furthermore, the pro-social fully isolated fish (*Figure 2B*: Fi (+S)), who exhibited an increase of freezes and reduced motility compared to control fish when viewing conspecifics, showed a similar activation to pro-social controls in the caudal hypothalamus, but increased activity in the dorsal preoptic area. Interestingly, the preoptic area was activated differently in pro-social and anti-social isolated fish, with only the dorsal preoptic area strongly activated in the pro-social group (*Figure 2B and D*: Fi (+S) vs Fi, p=0.04 vPa, p=0.002 dPa, Man-Whitney). These data suggest that long social isolation causes abnormal neural responses during viewing of social cues.

Furthermore, anti- and pro-social fish exposed to a brief isolation for only 48 hr prior to testing, showed similar functional activity changes to fully isolated fish, albeit less strong (*Figure 2B and D*: Pi (-S) vs Pi (nsc), p=0.18 dHc; p=0.28 vHc; p=0.04 vPa; p=0.04 dPa, Mann-Whitney; *Figure 2B and D*: Pi (+S) vs Pi (nsc), p=0.17 dHc; p=0.05 vHc; p=0.007 vPa; p=0.006 dPa). Together with the behavioural data, this finding supports the idea that short term isolation is enough to induce brain activity changes similar to those observed following complete isolation, and strikingly different than those observed in anti-social controls.

We were next interested in understanding why social isolation promotes social aversion instead of increasing the drive for social interaction. An important clue was found in the pattern of brain activity changes that were unique to isolated fish. When we directly compared the normalised *c-fos* functional brain maps of isolated and control fish that were not exposed to social cues during the assay (*Figure 3A*), we found a significant increase in two interesting areas; one associated with visual

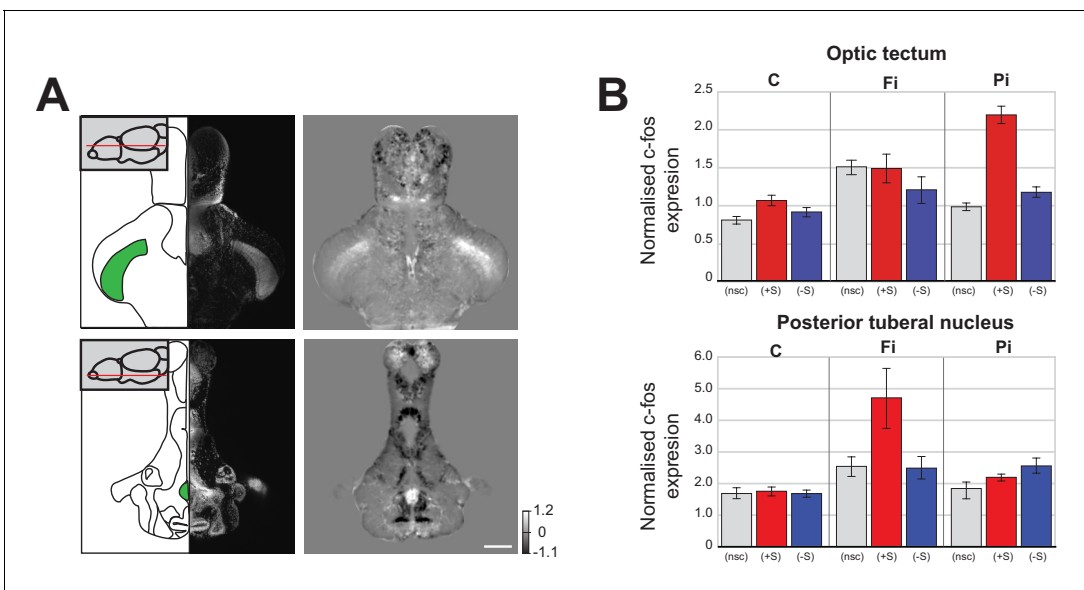

**Figure 3.** Changes in baseline brain activity following isolation. (**A**) Images of two areas that show strong c-fos activation in fully isolated fish independent of social stimuli (optic tectum and posterior tuberal nucleus (PTN)). Schematics of the horizontal sections and corresponding DAPI image are shown in the left panels. One plane of the 3D mask regions used for subsequent analysis is indicated (green). Images of Fully isolated fish c-fos neuronal activity, calculated as average differences between fully isolated (Fi) fish and normally raised fish without social cues (nsc) are shown in the right panels. Scale bar 200 μm. (**B**) Summary graphs showing the normalised c-fos expression in the optic tectum and PTN 3D masks for each experimental condition: non social cue (nsc), pro-social (+S) and anti-social (-S) for all the controls (C), fully isolated (Fi), and partially isolated (Pi) fish.

processing, the optic tectum, [*McDowell et al., 2004*]), and one involved in stress responses, the posterior tuberal nucleus (*Ziv et al., 2013*).

In pro-social control fish, viewing social cues resulted in a significant increase of neuronal activity in the optic tectum (*Figure 3B* top: C (+S) vs C (nsc), p=0.004 Mann-Whitney). However, in fully isolated fish, there was already increased neuronal activity in the optic tectum in the absence of social cues (*Figure 3B* top: Fi (nsc) vs C (nsc), p=0.0004, Mann-Whitney), suggesting that isolation increases visual sensitivity, as previously reported in humans (*Cacioppo et al., 2015*). This increased sensitivity of fully isolated fish not presented with social cues was weaker in partially isolated fish (*Figure 3B* top: Pi (nsc) vs C (nsc), p=0.03, Mann-Whitney). However, a much larger increase in tectal activity was observed when pro-social partially isolated fish viewed conspecifics, revealing that some visual sensitization had occurred (*Figure 3B* top: Pi (+S) vs C (+S), p=0.0002, Mann-Whitney). In addition, increased tectal activity was also present in both fully and partially isolated anti-social fish (*Figure 3B* top: Fi (-S) vs C (-S), p=0.048; Pi (-S) vs C (-S), p=0.005, Mann-Whitney), even though these fish largely avoided the chamber with visual access to conspecifics.

We also observed isolation-related activity increases in the posterior tuberal nucleus, an area associated with stress responses in zebrafish (*Wee et al., 2019*; *Ziv et al., 2013*). Full isolation caused a significant increase in posterior tuberal nucleus activity in the absence of social cues (*Figure 3B* bottom: Fi (nsc) vs C (nsc), p=0.015, Mann-Whitney) and in both anti-social and pro-social fish exposed to social cues (*Figure 3B* bottom: Fi (+S) vs C (+S), p=0.003; Fi (-S) vs C (-S), p=0.016, Mann-Whitney). Following partial isolation, posterior tuberal nucleus activity was not increased in the absence of social cues (*Figure 3B* bottom: Pi (nsc) vs C (nsc), p=0.29, Mann-Whitney), only slightly in pro-social fish (*Figure 3B* bottom: Pi (+S) vs C (+S), p=0.018), but significantly so in anti-social fish (*Figure 3B* bottom: Pi (-S) vs C (-S), p=0.0005).

Given these results from the optic tectum and posterior tuberal nucleus, we propose that isolation initially heightens sensitivity to social stimuli. However, when prolonged, this heightened sensitivity results in an increase of stress and anxiety levels during social viewing that leads to an aversion for social stimuli.

To test our hypothesis that reducing anxiety could reverse the anti-social behaviour observed in isolated zebrafish, we acutely treated control and partially isolated fish with Buspirone, an agonist of the auto-inhibitory 5HT$_{1A}$ receptor. The choice of isolation duration was motivated by the intermediate behavioural and functional phenotype of partial isolation relative to normal-rearing and full isolation, which would allow us to more easily detect both positive and negative impacts of treatment on sociality. The choice of Buspirone was supported by the changes in activity observed in the caudal hypothalamus of isolated fish, and by the fact that the caudal hypothalamus and the preoptic area strongly express Htr1ab receptors, one of the two orthologues of the 5HT$_{1A}$ receptor (*Norton et al., 2008*). Buspirone has been shown to reduce anxiety in humans, mice, and zebrafish (*Bencan et al., 2009*; *Lalonde and Strazielle, 2010*; *Lau et al., 2011*; *Patel and Hillard, 2006*). While it is not fully understood how Buspirone reduces anxiety, it has been shown to enhance social interaction of rats (*File and Seth, 2003*; *Gould et al., 2011*), sociability of zebrafish (*Barba-Escobedo and Gould, 2012*), and reduce social phobia in humans (*Schneier et al., 1993*; *van Vliet et al., 1997*). Its ability to counter the effects of social isolation in zebrafish has not been investigated.

We first tested the effects of acute exposure to Buspirone in control fish, and, as expected, we observed a small significant increase in social preference relative to untreated controls, however, a population of ~10% anti-social fish remained (*Figure 4—figure supplement 1*; C (no drug) vs C (30 µM), p=0.01, Mann-Whitney). We then treated partially isolated fish with 30 µM and 50 µM (*Figure 4—figure supplement 1*, n = 46, n = 72 fish) of Buspirone. Remarkably, the acute drug treatment was sufficient in both concentrations to reverse the anti-social phenotype caused by isolation (*Figure 4A*; Pi vs Pi (Buspirone 30 µM and 50 µM combined), p=2.56 e-05, Mann-Whitney).

When we then compared the time course of this phenotype reversal by computing the VPIs for each minute throughout the 15 min of the behavioural experiment (*Figure 4B*). We found that the isolated fish treated with Buspirone, while initially anti-social, would rapidly recover normal social preference behaviour within the first 5 min of exposure to social cues (*Figure 4B*: C vs Pi (Buspirone), p=0.016, first minute; p=0.37, fourth minute, Mann-Whitney). In contrast, the VPIs of untreated isolated fish remained significantly lower than controls throughout the entire session. We next compared the time course of movement activity (*Figure 4C*), and found that it generally increased quickly throughout the first 5 min of the social viewing session. Notably, the activity of isolated fish

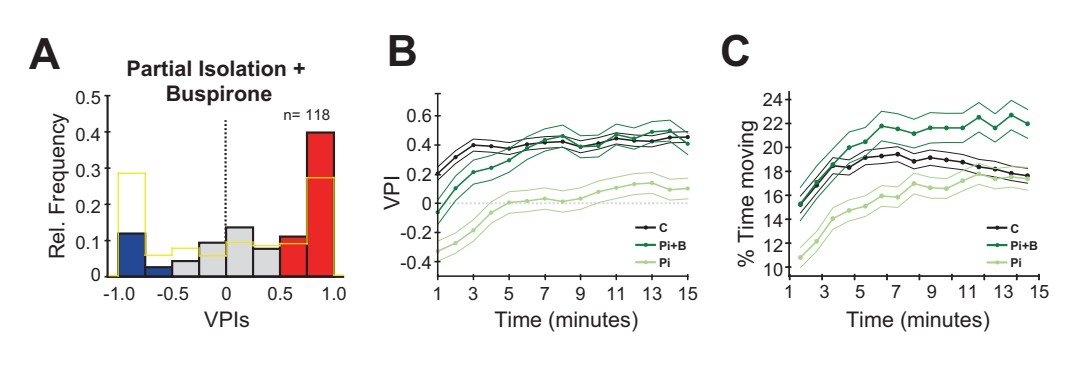

**Figure 4.** Buspirone rescues social preference in isolated fish. (A) Histogram of VPIs during the social cue period in partially isolated (Pi) fish treated with 30 μM and 50 μM of Buspirone (combined). For visual clarity, the bars are colored as in *Figure 1*. (B) VPI values calculated in one-minute time bins for controls (C, black line, n=380), partial isolated (Pi, blue line, n=157), and Pi treated with Buspirone (Pi+B, green line, n=118). Note how Buspirone treated fish recover normal social preference within the first 5 minutes. (C) Percentage of time moving calculated in one-minute bins for the same fish as B, thin lines indicate standard error.

The online version of this article includes the following figure supplement(s) for figure 4:

**Figure supplement 1.** Buspirone rescues social preference in isolated fish.

treated with Buspirone was already at the level of controls from the start of the social viewing session (*Figure 4B*: C vs Pi (Buspirone), p=0.31, first minute, Mann-Whitney), which suggests that the recovery of normal movement activity, possibly as a result of reduced anxiety, precedes the recovery of normal social preference. Therefore, Buspirone's impact on the rate of recovery of social preference indicates that it may do so by reducing anxiety, perhaps at the level of the preoptic and/or caudal hypothalamic area, allowing circuit plasticity to down-regulate the hypersensitivity to social stimuli acquired during the isolation period.

In summary, our results demonstrate that *lonely* fish, which have been isolated from social cues and show anti-social behaviour, have a completely different functional response to social stimuli than *loner* fish, anti-social fish found in the normal population. In addition, the functional changes caused by social deprivation are consistent with an increase in anxiety resulting from hyper-sensitization to social stimuli, similar to the effects of isolation on humans. We could reverse isolation's effects in zebrafish with an existing anxiolytic drug that acts on the monoaminergic system. Zebrafish will thus provide a powerful new tool for studying the impact of loneliness (isolation) on brain function and exploring different strategies for reducing, or even reversing, its harm.

# Materials and methods

## Key resources table

| Reagent type (species) or resource | Designation | Source or reference | Identifiers | Additional information |
|---|---|---|---|---|
| Antibody | Anti- digoxigenin-POD, sheep, polyclonal Fab fragments | Sigma-Aldrich, Rouche | Roche, Cat# 11207733910, RRID:AB_514500 | 1:3000 |
| Sequence-based reagent | cFos _F | This paper | PCR primers | CCGATACACTGCAAGCTGAA |
| Sequence-based reagent | cFos_R | This paper | PCR primers | ATTGCAGGGCTATGGAAGTG |
| Peptide, recombinant protein | Proteinase K | Sigma-Aldrich | Cat# P6556-10MG | 2 mg/ml |
| Commercial assay | TSA Plus Cyanine three system | Sigma-Aldrich, Perkin Elmer | Cat# NEL74401KT | Dilution 1:50 |

*Continued on next page*

*Continued*

| Reagent type (species) or resource | Designation | Source or reference | Identifiers | Additional information |
|---|---|---|---|---|
| Chemical compound, drug | Buspirone hydrochloride | Sigma-Aldrich | Cat# B7148-1G | 30 uM and 50 uM |
| Software, algorithm | Anaconda, Spyder | Anaconda (https://www.anaconda.com/) | Spyder, RRID:SCR_017585 | Version 4.0.1 |
| Software | ImageJ | NIH (http://imagej.nih.gov/ij/) | RRID:SCR_003070 | |
| Software | ANTs- Advanced Normalisation Tools | http://stnava.github.io/ANTs/ | RRID:SCR_004757 | Version 2.1.0 |
| Other | DAPI staining | Sigma-Aldrich | Cat# D9564-10MG | 1 mg/ml |
| Other | Slc6a4b RNA probe | *Norton et al., 2008* | | |
| Other | DAT RNA probe | *Filippi et al., 2010* | | |
| Other | Th1 RNA Probe | *Filippi et al., 2010* | | |
| Other | Th2 RNA probe | *Filippi et al., 2010* | | |

## Animals

AB strain zebrafish maintenance and breeding was performed at 28.5C with a 14 hr:10 hr light-dark cycle. Isolated fish were housed in custom chambers (length = 15 cm, width = 5 cm, height = 10 cm) made of opaque white acrylic with translucent lids, either from fertilization (full isolation) or for 48 hr prior to the behavioural experiment (partial isolation). All experiments were performed according to protocols approved by local ethical committee (AWERB Bloomsbury Campus UCL) and the UK Home Office.

## Behavioural assay and analysis

Experimental details and image acquisition were performed as described previously (*Dreosti et al., 2015*). Fish were positioned in custom-built behavioural arenas (*Figure 1D*) made of white acrylic, and illuminated with visible light using a laser light projector (Microvision, ShowwX+, US). The videography system comprised a high-speed camera (Flea3, PointGrey, CA), an infrared light (Advanced Illumination, US, 880 nm), an IR filter (R70, Hoya, JP), and a vari-focal lens (Fujinon, JP). Experiments were recorded using custom written workflows in Bonsai (*Langen et al., 2015*). Test fish were positioned in the main C-shape compartment of the arena by pipetting, and left for 15 min to acclimate. A social cue, two fish of same age and similar size, was then introduced into one of the two adjacent chambers randomly. Test fish could see the social cue through a glass window. Each fish was run only once in the behavioural assay.

Images were analysed using custom written computer vision scripts in Python based on OpenCV (https://www.dreo-sci.com/resources/). Each frame was cropped, background subtracted, and thresholded. The centroid, position, orientation, and per frame motions of the test fish were identified, and stored in a CSV file. All videos have been saved with H.264 compression for subsequent offline analysis, and are available upon request.The source code can be downloaded at http://www.dreo-sci.com/resources/.

The visual preference index (VPI) was calculated by subtracting the number of frames in which the fish was located on the side of the arena nearest the social stimulus (Social cue (SC) side in *Figure 1B*) by the number of frames located on the opposite side of the arena (nsc (No SC) side). This difference was then divided by the total number of frames recorded [VPI = (SC – No SC)/Total frames]. The percent time moving was calculated by counting each frame with detectable changes in the fish image relative to the previous frame (i.e. motion), and dividing by the total number of frames. The percent time freezing was calculated by detecting contiguous sequences without motion longer than 3 s, counting all frames that are part of such sequences, and dividing by the total number of frames.

## Whole mount in situ hybridisation

Fluorescent in situ hybridizations using digoxigenin-labelled *c-fos* were performed on dissected juvenile zebrafish with few modification to the original method (*Brend and Holley, 2009*). After overnight fixation in 4% PFA, protein K treatment (2 mg/ml 20 min of incubation), inactivation of endogenous peroxidase with $H_2O_2$ (22% v/v for 30 min at room temperature), additional fixation (30 min at room temperature) and 3 hr of incubation with the hybridisation buffer, fish were incubated with the *c-fos* probe (courtesy from Ricardo N. Silva (Forward CCGATACACTGCAAGCTGAA and Reverse ATTGCAGGGCTATGGAAGTG), or with dopamine transporter (DAT), tyrosine hydroxylase 1 (Th1), tyrosine hydroxylase (Th2) (*Filippi et al., 2010*), or the 5-HT transporter, solute carrier family 6 member 4b (Slc6a4b) probes (*Norton et al., 2008*). C-*fos*, *DAT* and *Slc6a4b* probes were detected with anti-Digoxigenin-POD, Fab fragments (Roche, 1:3000) and TSA Plus Cyanine 3 System (Perkin Elmer, 1:50). Nuclear staining was obtained using DAPI (Sigma-Aldrich, 1: 500). Fish were then mounted for imaging in low melting point agarose (2.5% Agarose, 0.8% glycerol, PBS-Tween) and imaged.

## Imaging and registration

A custom built two-photon microscope (INSS) was used for image acquisition of whole-brain in situs. Both DAPI and Cy3 Images were collected with a 10x objective (Olympus, W Plan-Apochromat 10x/0.5 M27 75 mm) using a 'Chameleon' titanium–sapphire laser tuned to 1030 nm (Coherent Inc, Santa Clara, CA, US) and controlled using custom written software in LabView. Registration of in-situ images was performed using ANTs (Advanced Normalisation Tools) version 2.1.0 running on the UCL Legion compute cluster. Images were down-sampled to 512*512 and parameters were slightly modified from *Marquart et al. (2017)* fixed registration:

```
antsRegistration -d 3 –float 1 -o [Registered_Image_, Registered_Image _warped.
nii.gz] –interpolation WelchWindowedSinc –use-histogram-matching 0 r [referen-
ce_Image, Registered_Image,1] -t rigid[0.1] -m MI[reference_Image, Registered_-
Image _0.nii,1,32,Regular,0.25] -c [1000 × 500×250 × 100,1e-8,10] –shrink-
factors 12 × 8×4 × 2 s 4 × 3×2 × 1 t Affine[0.1] -m MI[reference_Image, Regis-
tered_Image,1,32,Regular,0.25] -c [1000 × 500×250 × 100,1e-8,10] –shrink-fac-
tors 12 × 8×4 × 2 s 4 × 3×2 × 1 t SyN[0.1,6,0] –m CC[reference_Image,
Registered_Image _0.nii,1,2] -c [1000 × 500×500x250 × 100,1e-7,10] –shrink-fac-
tors 12 × 8×4x2 × 1 s 4 × 3×2x1 × 0
antsApplyTransforms -d 3 v 0 –float -n WelchWindowedSinc -i Registered_Image _1.
nii -r reference_Image -o Registered_Image _warped_red.nii.gz -t Registered_-
Image _1Warp.nii.gz -t Registered_Image _0GenericAffine.mat
```

## Intensity normalisation

The registered image stacks were then normalised to adjust for intensity variations between imaging sessions caused by a variety of sources (staining efficiency, laser power fluctuations, light detector sensitivity, etc.). Normalisation was accomplished by computing an intensity histogram for each fish brain's volume (with 10000 discrete intensity bins spanning the range −4000.0 to 70000.0) for all 512*512*273 voxels. The minimum value bin (with at least 100 voxels) was used as the bias offset, and subtracted from all voxel values. The mode value, minus the bias, provided a robust estimate of the background/baseline fluorescence and was thus used to normalise voxel values for the entire volume. Therefore, after normalisation, an intensity value of 1 reflected the background level while two indicates fluorescence level that is twice the background, and so on. Histogram normalisation was performed for each individual fish's brain volume prior to any region or voxel-based analysis.

*Figures 2B* and *3A* Reconstruction of cross section images were obtained by using the Fiji 'Volume viewer' plugin. Schematics of cross- and horizontal-section were obtained by using the 'Neuroanatomy of the zebrafish brain'.

*Figures 2D* Percentages of *c-fos* activation were calculated for each of the six different areas highlighted in *Figures 2B* and *3A*, using custom written Python functions, in the following way. A 3D mask for each area was generated by using the 'Segmentation Editor' plugin Fiji (https://imagej.net/

Segmentation_Editor). *C-fos* percentage values for each condition (C (+S), C (-S), Fi (-S), Pi (-S)) were obtained by subtracting and then dividing each *c-fos* average value of the mask by the basal *c-fos* average value calculated in control fish No Social Cue.

## Statistics

Statistical analysis was performed using Python scipy stats libraries. Since VPI, percent time moving/ freezing, and *c-fos* activity distributions were generally not normally distributed, we used the non-parametric Mann-Whitney U-test of independent samples for hypothesis testing throughout the manuscript.

## Drug treatment

Juvenile fish were treated with 30 µM or 50 µM Buspirone (Buspirone HCl, Sigma) for 10 min prior the experiment. After washing, fish were run through the behavioural assay. Each fish was used only once.

## Data availability

All the images, video, protocols, analysis scripts, and data that support the findings of this study are available from this website (http://www.dreo-sci.com/resources/), or our GitHub repository (https://github.com/Dreosti-Lab/Lonely_Fish_2020; *Dreosti, 2020*; copy archived at https://github.com/elifesciences-publications/Lonely_Fish_2020), or from the corresponding author upon request.

## Acknowledgements

The authors acknowledge Steve Wilson for providing lab resources. Ricardo Neto Silva for the *c-fos* probe. Jason Rihel for some of the reagents. Wolfgang Driver for the dopaminergic probes and William Norton for the serotoninergic probes. UCL fish facility team for fish care/husbandry. This work was supported by the Wellcome Trust Grant Ref 202465/Z/16/Z.

## Additional information

### Funding

| Funder | Grant reference number | Author |
|---|---|---|
| Wellcome | 202465/Z/16/Z | Elena Dreosti |
| Gatsby Charitable Foundation | 090843/F/09/Z | Adam Raymond Kampff |

The funders had no role in study design, data collection and interpretation, or the decision to submit the work for publication.

### Author contributions

Hande Tunbak, Conceptualization, Data curation, Formal analysis, Validation, Investigation, Visualization, Methodology, Writing - review and editing; Mireya Vazquez-Prada, Data curation; Thomas Michael Ryan, Software, Writing - review and editing; Adam Raymond Kampff, Software, Formal analysis, Visualization, Writing - review and editing; Elena Dreosti, Conceptualization, Resources, Data curation, Software, Formal analysis, Supervision, Funding acquisition, Validation, Investigation, Visualization, Methodology, Writing - original draft, Project administration, Writing - review and editing

### Author ORCIDs

Hande Tunbak (iD) https://orcid.org/0000-0003-3180-1401
Mireya Vazquez-Prada (iD) https://orcid.org/0000-0001-7964-7576
Thomas Michael Ryan (iD) https://orcid.org/0000-0001-9469-4135
Adam Raymond Kampff (iD) https://orcid.org/0000-0003-3079-019X
Elena Dreosti (iD) https://orcid.org/0000-0002-6738-7057

## Ethics

Animal experimentation: All experiments were performed according to protocols approved by local ethical committee (AWERB Bloomsbury Campus UCL) and the UK Home Office. PAE2ECA7E.

## Decision letter and Author response

Decision letter https://doi.org/10.7554/eLife.55863.sa1
Author response https://doi.org/10.7554/eLife.55863.sa2

## Additional files

### Supplementary files

• Transparent reporting form

### Data availability

All the images, video, protocols, analysis scripts, and data that support the findings of this study are available on our website (www.dreo-sci.com/resources) and GitHub repository (https://github/Dreosti-Lab/Social_Zebrafish), or from the corresponding author upon request.

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
