## [Decision Letter]

**Acceptance summary:**

I agree with you that in these times of widespread isolation, the questions addressed in this report, and the distinction between a self chosen loner mode and an extrinsically forced loneliness are ever so relevant.

**Decision letter after peer review:**

Thank you for submitting your article "Whole brain mapping reveals abnormal activity in socially isolated zebrafish" for consideration by *eLife*. Your article has been reviewed by four peer reviewers, including Peggy Mason as the Reviewing Editor and Reviewer #1, and the evaluation has been overseen by Catherine Dulac as the Senior Editor.

The reviewers discussed the reviews with one another and the Reviewing Editor has drafted this decision to help you prepare a revised submission.

The most important point is that we would like for you to test whether the hypothalamic neurons are serotonergic using a method of your choice. As we discussed this manuscript just a week or two ago, we felt that the most important point was that we wanted you to test whether the hypothalamic neurons are serotonergic using a method of your choice. I continued with the explicit statement that "Acceptance will not be contingent on a specific answer – yes they are as you suspect or no they aren't. Either finding is fine. Obviously those two possibilities will require very different interpretations and discussions. It may also be that you choose to submit your manuscript, as is and without these added experiments, to a different journal. Or to resubmit with these experiments to *eLife*. Any of these courses of action are fine."

However, the world has changed since our editorial discussions took place. I am assuming that you, along with the bulk of scientists, do not have access to their research laboratories. I do not want to hold up your publication indefinitely as the world navigates the pandemic. I therefore offer you the option of proceeding with purely textual changes and explicitly withdraw the requirement of additional experiments. Of course, should you have those experiments in hand, we welcome their inclusion in the revised manuscript that we look forward to receiving.

A number of other points that stand out from the many suggestions that you will find below.

1) Introduction tightened up to foreshadow the key takeaway message as suggested by reviewer 4.

2) A clear indication of what the N counts, as reviewer 2 asks for. Also please explicate the full method of behavioural assay and analysis rather than referring to another paper.

3) It appears that immobility is used as a metric for "anxiety." If so, this should be quantified and immobility should take the place of anxiety in the results. Moreover. immobility can also signal fear- or depression-like affect. Thus, the rhetoric of the discussion should be toned down to recognize this ambiguity. It is noteworthy in this regard that buspirone has atypical antidepressant as well as anxiolytic effects.

4) Please explain why Busp was tested on partially isolated rather than total isolates.

One final point, there was disagreement among the reviewers regarding the use of the terms lonely and loner. I am going to take editor's prerogative and call this one on the side of keeping those terms and even extending their use. But to the dissenters' point, please add some justification and caveats to your use of these terms.

Reviewer #1:

This is a very interesting and compelling study showing that the lack of a preference for being near other fish that is typical is not present after social isolation. But the exciting piece is that the cfos activation is different in control raised animals that avoid other fish and in isolated fish that show the same avoidance. The authors suggest use the terms lonely and loner for isolated and control fish that fail to prefer other fish.

My only major concern is that the "anxiety" behavior is talked about but not shown or quantified. Figure 1D is referenced but it is not obvious to me how that figure of location shows me the amount of freezing. The solid black areas could be freezing or repetitive swimming in the same area – I can't tell which. And I would be happier if they called it freezing if that is in fact how they measured it.

Reviewer #2:

This paper utilizes zebrafish in order to study the effects of chronic social isolation on neural responses. The authors note that within a normal zebrafish population, there exists some variability in the social preference of the animals, such that while most animals display a strong tendency to shoal with a visually presented conspecific, there is a certain percentage that will not display this behavior (or display it to a very limited extent), a behavior that is reminiscent of what is observed in chronically isolated animals, which also tend to avoid social interaction. The authors then attempt to compare the neural response in particular the optic tectum and regions of the ventral midbrain (Preoptic nucleus of the hypothalamus, Posterior tuberal nucleus, and caudal hypothalamus) to visually presented social stimuli, and draw a distinction between these responses in chronically isolated fish, acutely isolated fish, and the response of a "normal" socially avoidant fish. They show that the neural responses in these brain areas differ as a function of isolation, and thereby conclude that the social avoidance behavior triggered by developmental isolation is not similar to the social avoidance found naturally within the population. Lastly, they rescue the anti-social phenotype by utilizing the anxiolytic drug Buspirone, both in partially isolated and non isolated fish, thus putatively showing that anti-social behavior is related to stress.

1) General comment: The authors report highly significant p-values for their findings (on the order of ^-8), this is to be expected given the reported group sizes. Are these group sizes actually from one experimental repeat (e.g n = 380 for non-isolated fish), or have they pooled together several repeats? If so then it would be more appropriate to analyze the repeated experiments by some other method (e.g. a 2-way-ANOVA or Generalized Linear Model with factors: "experiment number" and "treatment"), rather than pool them and utilize a single test for all of the data.

2) Third paragraph: "we found that fish raised in isolation were significantly less active than their normally raised siblings during the acclimation session (Figure 1B: C vs Fi, p=9.0e-6; C vs Pi, p=2.8e-9 Mann-Whitney). However, when divided into groups with similar social preferences, pro-social isolated fish were nearly as active as pro-social controls (Figure 1C right: C (+S) vs Fi (+S), p=0.02 Mann-Whitney; C (+S) vs Pi (+S) p=0.14 Mann-Whitney"

These statements are confusing/misleading. The authors did not compare the activity levels of pro-social, non-social, and anti-social fish in the same paradigm as they did in the first sentence (namely, the acclimation phase). They measure it during the social exposure phase (Figure 1C, legend), thus they actually show a differential response to the social cue in terms of overall motility, but not a change in baseline activity as a function of sociality levels per se. I agree that it would have been interesting to compare, post-hoc, the baseline activity levels of the pro\non\anti-social fish (i.e. classify the fish based on their score in the social assay, and find their baseline activity during acclimation, and compare between the groups), but this is not the data presented herein , and the phrasing ought to be changed to reflect that, or the correct comparison presented in the figure.

3)."…which were both not exposed to social cues during the assay (Figure 3A), we found a significant increase in areas associated with visual processing the optic tectum, (McDowell et al., 2004) and stress responses the posterior tuberal nucleus, PTN"

As far as I can tell, the authors did not attempt to assess the identity of the PTN activated neurons and therefore this statement may be a bit strong. For example, the PTN is also the major area of fish dopaminergic neurons associated with reward as well GABA-ergic neurons.

4) Paragraph four-six: What was the data normalized to in these experiments. In the preceding experiment (Figure 2) where the authors look at the response of the social areas, they compare the socially exposed to fish to socially nonexposed fish, but this cannot be the case here as the socially unexposed fish are part of the analysis (and their activity levels do not sum up to 1 on the normalized activity graph).

5) "…and social phobia in humans (Schneier et al., 1993; van Vliet et al., 1997)" should read: "reduces social phobia", not enhances.

6) "Buspirone's impact on the rate of recovery of social preference suggests it may be reducing anxiety by regulating circuit plasticity, perhaps by promoting down-regulation of the hypersensitivity acquired during the isolation period"

The authors previously quantified anxiety-like behaviors via measures of mobility (% time spent moving). Therefore, it would be appropriate to include a measure of time spent moving per minute in order to strengthen this claim. As it stands, this claim is an interesting hypothesis, but not quite shown by the data.

7) Three comments on the Buspirone rescue experiment:

a) Why did the authors only perform this experiment on partially isolated fish, and not fully isolated ones? Given the differences in behavior and brain activity between these two models it is not at all clear that the neural mechanism which drives their anti-social behavior is the same, and therefore full isolates may not be amenable to treatment by Buspirone. This would be an interesting and relevant addition to the paper.

b) Buspirone rescued anti-social behavior in Control (non-isolated) fish, and also in partially isolated fish. The authors main point in the paper is that the neural mechanisms which drive anti-social behavior following isolation are caused by increased anxiety, and are distinct from those that drive it in the general population. However, since Buspirone rescued anti-social behavior also in controls, does this not mean that anti-social controls and anti-social partially isolated animals share some neural phenotype?

c) If possible, it would be instructive to perform neural activity imaging on Buspirone treated animals, and in particular, show reduced activity in the optic tectum and PTN, as the authors claim that hyperactivity in these regions is anxiogenic and drives the anti-social behavior of isolates. Thus, if the authors hypothesis is correct, it would be expected that Buspirone treatment, which rescues the behavior, would also reduce activity in these areas.

Reviewer #3:

This study identifies brains areas that control social behavioural responses towards conspecifics in socially isolated and normally reared zebrafish. I am excited about this work because it opens up the possibility to understand the neurotransmitters and brain areas that control social interactions. The study is short and the data that are presented are clear. In particular, the differences in cfos expression between animals is very impressive.

This study would be improved significantly by directly measuring 5-HT levels in these animals (see below). In the absence of this, some of the conclusions are too strong. I also have some suggestions for improving the Discussion section and tidying up the writing.

1) This research could be improved by directly measuring 5-HT levels in the hypothalamus of isolated or socially-reared fish. This could be done by HPLC. The caudal hypothalamus contains a number of different neuron types, so IEG activity here could reflect activation of several neurotransmitter systems. In the absence of HPLC data, you need to consider other neurotransmitters that could also contribute to this phenotype.

2) The whole brain imaging of cfos is neat, but it loses any cellular resolution. The findings would be strengthened by making sections of the brain and taking higher magnification images. This could include co-labelling with specific markers for dopamine and 5-HT neurons.

3) The Abstract needs rewriting because the tense used changes throughout.

4) The second paragraph of the Introduction (describing isolation and the experimental setup) is not that clear, and can be rewritten to make it easier to follow.

5) There are a large number of acronyms for brain areas used. It might be better to replace all of these with the full names of the brain areas. This would also improve consistency – for example, you write preoptic (area) but not POA. It will also help to distinguish brain areas from treatment groups (nsc, Pi, Fi) etc.

6) Similarly please check name usage throughout – cf preoptic area, preoptic, pre-optic.

7) I missed the link to sertb expression in the hypothalamus. I would highlight this more clearly, particularly because you use buspirone to rescue behaviour in later experiments.

8) Some of the discussion points can be improved. There are already studies linking the POA to social behaviour in zebrafish, and these should be included here. It is possible that arginine vasopressin plays an important role in this behaviour. Linking the central hypothalamus to the control of social behaviour is also interesting; the function of sertb positive neurons is well studied. Again, this can be highlighted here.

9) It is not clear whether the caudal hypothalamus identified by cfos staining here is the same as the ventromedial hypothalamus described by O'Connell and Hoffmann.

10) Final paragraph of the Discussion I would remove the terms "lonely" and "loner". They are not necessary here and seem anthropomorphic.

Reviewer #4:

This manuscript describes a series of studies exploring the impacts of social isolation on zebrafish behavior and neural activity. In particular, the authors compare "loner" fish who by choice spend little time socializing with "lonely" fish who are experimentally isolated and discover that while behavior is relatively similar, neural activity is fundamentally different, the latter being rescued by reducing serotonin. In general, I like the paper a lot; it represents a significant effort by the authors and suggests that despite behavioral similarities, all isolation is not the same. This could profoundly shape how we think about sociality in general, and the impacts of isolation in particular. However, there are some aspects of the manuscript that need to be clarified.

Substantive concerns:

1) One of my big concerns is that the initial paragraph of manuscript does not do a good job of setting up the paper. The first paragraph needs to prepare the reader for the "lonely" vs "loner" fish comparison and explain why it matters. Also, the partial isolation fish are introduced in the second paragraph with no explanation of why they are important. I figured all of this out by the end of the manuscript, but the paper will be much stronger if the reader knows in the first paragraph what to expect. Specifically, I would add conceptual overview that sets the reader up for all of the comparisons and emphasizes why each is needed and important.

2) The only places I can find the sample size are in Figure 1 and the paragraph describing the Buspirone treatment. Please provide details on sample size and whether it was sufficiently powered, either throughout or in the Materials and methods section.

---

## [Author Response]

A number of other points that stand out from the many suggestions that you will find below.1) Introduction tightened up to foreshadow the key takeaway message as suggested by reviewer 4.

We agree that the Introduction, and especially the first paragraph, did not clearly emphasise the main findings of the paper. We have now added a more general overview of the study’s goals and outcomes, introduced the concept of “lonely” and “loner”, and explained why we decided to use both partial and full isolation.

2) A clear indication of what the N counts, as reviewer 2 asks for. Also please explicate the full method of behavioural assay and analysis rather than referring to another paper.

Thank you for pointing this out. We now include N counts for every experiment reported, either in the figure directly or in the corresponding legend. As suggested, we have modified the Materials and methods section to include a complete explanation of the behavioural assay and analysis as opposed to referring to previous work. As requested by reviewer 2, we have also specified that each fish has been tested only once, i.e. 380 individual control fish were tested in the social assay, and therefore all the experiments are single trials and it is appropriate to use the Mann-Whitney statistical test as opposed to ANOVA.

3) It appears that immobility is used as a metric for "anxiety." If so, this should be quantified and immobility should take the place of anxiety in the results. Moreover. immobility can also signal fear- or depression-like affect. Thus, the rhetoric of the discussion should be toned down to recognize this ambiguity. It is noteworthy in this regard that buspirone has atypical antidepressant as well as anxiolytic effects.

Thanks for highlighting this ambiguity. We agree that the measure we used in the paper, percentage of time moving, could underlie other behavioural phenotypes apart from anxiety, such as depression or fear, and we have modified our Discussion accordingly. We have also now included a supplementary Figure (Figure 1—figure supplement 1C-D) where we quantify the percentage of time spent freezing during acclimation and social cue exposure. This parameter reports the percentage of time that fish remain motionless for 3 seconds and longer. The data show that fully and partially isolated fish show a significant increase in “freezes” during acclimation and social cue exposure, although fully isolated fish show a larger increase (Figure 1—figure supplement 1C-D, left). Furthermore, when we look at the anti-social fish during social cue exposure (Figure 1—figure supplement 1D right, blue dots), each experimental group (controls, fully isolated, and partially isolated), has a similar percentage of time freezing, supporting our claim that freezing is a hallmark of anti-social behaviour for both “loner” and “lonely” fish.

Finally, it is true that Buspirone has also been used as an atypical antidepressant, however, only as adjunctive treatment in patients with generalized anxiety disorders and depression. This is very interesting. We believe that there is still much to learn about the action of Buspirone and the interaction between anxiety and depression, and we feel that zebrafish will provide an excellent model for further study. We have therefore improved our discussion of Buspirone throughout the manuscript, both to motivate its use in our experiments and to suggest future directions.

4) Please explain why Busp was tested on partially isolated rather than total isolates.

We agree that this point needs to be better explained in the paper, thank you for highlighting this. There are several reasons why we opted for partially versus fully isolated fish to perform the drug experiments. First of all, partial isolation of only 48 hours instead of 3 weeks already induces a profound behavioural change in social preference behaviour (Figure 1A) similar to full isolation. Secondly, the functional activity observed in the presence (Figure 2B) or not (Figure 3A-B) of social cues is very similar between fully and partially isolated fish. However, in both behaviour and neural activity, the impact of partial isolation is less severe than full isolation. We therefore decided that partial isolation represented an “intermediate” phenotype that would allow us to detect both positive and negative impacts on sociality for a given treatment. This has now been explained in the manuscript.

One final point, there was disagreement among the reviewers regarding the use of the terms lonely and loner. I am going to take editor's prerogative and call this one on the side of keeping those terms and even extending their use. But to the dissenters' point, please add some justification and caveats to your use of these terms.

Thank you for this decision. We also believe that the terms are compelling and we did include them in the title of our first submission. Therefore, we have reintroduced the terms in the title, explained and justified there use in the Introduction, and used them throughout the manuscript where appropriate.

Reviewer #1:This is a very interesting and compelling study showing that the lack of a preference for being near other fish that is typical is not present after social isolation. But the exciting piece is that the cfos activation is different in control raised animals that avoid other fish and in isolated fish that show the same avoidance. The authors suggest use the terms lonely and loner for isolated and control fish that fail to prefer other fish.My only major concern is that the "anxiety" behavior is talked about but not shown or quantified. Figure 1D is referenced but it is not obvious to me how that figure of location shows me the amount of freezing. The solid black areas could be freezing or repetitive swimming in the same area – I can't tell which. And I would be happier if they called it freezing if that is in fact how they measured it.

We agree that the qualitative presentation of “freezing” behaviour in Figure 1D and the supplemental video were insufficient to demonstrate that anti-social fish exhibit anxiety-like behaviour. Therefore, we have now quantified the percentage of time spent freezing during acclimation and social cue period and included this analysis in Figure 1—figure supplement 1C-D. The data show that both fully and partially isolated fish have a significant increase in freezes during the acclimation period, suggesting a heightened level of general anxiety. This increased freezing persists during social cue exposure, but is largest in fully isolated fish. We believe this additional analysis greatly supports our claim that isolation increases anxiety-like freezing behaviour. Thank you very much for the suggestion.

Reviewer #2:This paper utilizes zebrafish in order to study the effects of chronic social isolation on neural responses. The authors note that within a normal zebrafish population, there exists some variability in the social preference of the animals, such that while most animals display a strong tendency to shoal with a visually presented conspecific, there is a certain percentage that will not display this behavior (or display it to a very limited extent), a behavior that is reminiscent of what is observed in chronically isolated animals, which also tend to avoid social interaction. The authors then attempt to compare the neural response in particular the optic tectum and regions of the ventral midbrain (Preoptic nucleus of the hypothalamus, Posterior tuberal nucleus, and caudal hypothalamus) to visually presented social stimuli, and draw a distinction between these responses in chronically isolated fish, acutely isolated fish, and the response of a "normal" socially avoidant fish. They show that the neural responses in these brain areas differ as a function of isolation, and thereby conclude that the social avoidance behavior triggered by developmental isolation is not similar to the social avoidance found naturally within the population. Lastly, they rescue the anti-social phenotype by utilizing the anxiolytic drug Buspirone, both in partially isolated and non isolated fish, thus putatively showing that anti-social behavior is related to stress.1) General comment: The authors report highly significant p-values for their findings (on the order of ^-8), this is to be expected given the reported group sizes. Are these group sizes actually from one experimental repeat (e.g n = 380 for non-isolated fish), or have they pooled together several repeats? If so then it would be more appropriate to analyze the repeated experiments by some other method (e.g. a 2-way-ANOVA or Generalized Linear Model with factors: "experiment number" and "treatment"), rather than pool them and utilize a single test for all of the data.

Thank you for pointing this out. The experiments are all single experimental repeats (i.e. 380 individual non-isolated fish were tested once in the social preference assay). We have now described this more clearly in the Materials and methods section.

2) Third paragraph: "we found that fish raised in isolation were significantly less active than their normally raised siblings during the acclimation session (Figure 1B: C vs Fi, p=9.0e-6; C vs Pi, p=2.8e-9 Mann-Whitney). However, when divided into groups with similar social preferences, pro-social isolated fish were nearly as active as pro-social controls (Figure 1C right: C (+S) vs Fi (+S), p=0.02 Mann-Whitney; C (+S) vs Pi (+S) p=0.14 Mann-Whitney"These statements are confusing/misleading. The authors did not compare the activity levels of pro-social, non-social, and anti-social fish in the same paradigm as they did in the first sentence (namely, the acclimation phase). They measure it during the social exposure phase (Figure 1C, legend), thus they actually show a differential response to the social cue in terms of overall motility, but not a change in baseline activity as a function of sociality levels per se. I agree that it would have been interesting to compare, post-hoc, the baseline activity levels of the pro\non\anti-social fish (i.e. classify the fish based on their score in the social assay, and find their baseline activity during acclimation, and compare between the groups), but this is not the data presented herein , and the phrasing ought to be changed to reflect that, or the correct comparison presented in the figure.

This is a really good point. Thank you for highlighting our confusing presentation and description. We have now changed Figure 1C and replaced the swarm plots of anti-social and pro-social fish during the social exposure with the ones during the acclimation period (that were classified based on their subsequent performance in the social cue period). The new data support our argument that the baseline (i.e. without social cues) mobility of anti-social control, fully, and partially isolated fish is very similar during the acclimation period. This reduced mobility persists during the social cue phase, as we originally reported in the main figure. We have moved the swarm plots of mobility during the social cue period to the Figure 1—figure supplement 1.

3) "…which were both not exposed to social cues during the assay (Figure 3A), we found a significant increase in areas associated with visual processing the optic tectum, (McDowell et al., 2004) and stress responses the posterior tuberal nucleus, PTN"As far as I can tell, the authors did not attempt to assess the identity of the PTN activated neurons and therefore this statement may be a bit strong. For example, the PTN is also the major area of fish dopaminergic neurons associated with reward as well GABA-ergic neurons.

Thank you for helping us to clarify this point. We have recently begun to perform *in situs* for dopaminergic, serotoninergic and acetylcholine markers in juvenile fish. These preliminary data show that the area activated is the PTN, however, the dopaminergic neurons are not the ones that we see activated. These dopaminergic neurons seem to surround the activated area shown in Figure 3A bottom. We do not know yet, what type of markers these neurons express. PTN neurons have been shown to express GABA (gad) and can also coexpress (vglut2) in addition to dopamine (Th1). We are planning to continue the characterization of the PTN neurons as soon as possible, but for now, we have improved our discussion of the PTN and other identified areas, throughout the manuscript to reflect these possible alternate roles.

4) Paragraph four-six: What was the data normalized to in these experiments. In the preceding experiment (Figure 2) where the authors look at the response of the social areas, they compare the socially exposed to fish to socially nonexposed fish, but this cannot be the case here as the socially unexposed fish are part of the analysis (and their activity levels do not sum up to 1 on the normalized activity graph).

Thank you for highlighting this confusion. All individual brain maps were first processed using an intensity histogram normalization, as described in the Materials and methods. As you point out, in Figure 2 we present the data as “difference images” to highlight relative changes in response to social cue exposure. In Figure 3, however, we compare between groups (i.e. controls vs. isolation) and identify *c-fos* activity changes that occur regardless of social cue presentation, and therefore we report the absolute (histogram normalized) *c-fos* expression in different regions. We have now emphasized the difference in presentation for Figure 2 and 3 in the manuscript text to highlight this for the reader.

5) "…and social phobia in humans (Schneier et al., 1993; van Vliet et al., 1997)" should read: "reduces social phobia", not enhances.

Thank you for finding this error. We have changed the text accordingly.

6) "Buspirone's impact on the rate of recovery of social preference suggests it may be reducing anxiety by regulating circuit plasticity, perhaps by promoting down-regulation of the hypersensitivity acquired during the isolation period"The authors previously quantified anxiety-like behaviors via measures of mobility (% time spent moving). Therefore, it would be appropriate to include a measure of time spent moving per minute in order to strengthen this claim. As it stands, this claim is an interesting hypothesis, but not quite shown by the data.

Thank you for suggesting this important analysis. We have now quantified the percentage of time moving in one minute bins for each treatment group presented in Figure 4 A-B and included a new panel in Figure 4 (Figure 4C). The results show that Buspirone treatment increases the mobility of partially isolated fish, which reach control levels within the first minute of the social assay. Therefore, these new data, together with the VPI results (Figure 4B), supports the argument that Buspirone reduces anxiety/increases mobility and that this happens prior to a recovery of social preference. Very interesting and a great addition to the study. Thanks again for the suggestion.

7) Three comments on the Buspirone rescue experiment:a) Why did the authors only perform this experiment on partially isolated fish, and not fully isolated ones? Given the differences in behavior and brain activity between these two models it is not at all clear that the neural mechanism which drives their anti-social behavior is the same, and therefore full isolates may not be amenable to treatment by Buspirone. This would be an interesting and relevant addition to the paper.

Thanks for allowing us to clarify this decision. In addition to the reply to the Editor’s comment above, we would like to say that we are (or we were prior to lockdown) currently pursuing experiments testing Buspirone treatment on fully-isolated fish. However, given that we have a limited number of isolation tanks, and that these tanks are entirely used by one fish for the full three week period of isolation, then these full isolation experiments are much lower throughput. Therefore, given that brief periods of isolation are likely much more common than complete lack of social interaction from birth, we decided to start with partial isolation fish to test possible treatments/rescues. However, we agree, that full (or at least longer) isolation tests need to follow for any identified treatment candidates.

b) Buspirone rescued anti-social behavior in Control (non-isolated) fish, and also in partially isolated fish. The authors main point in the paper is that the neural mechanisms which drive anti-social behavior following isolation are caused by increased anxiety, and are distinct from those that drive it in the general population. However, since Buspirone rescued anti-social behavior also in controls, does this not mean that anti-social controls and anti-social partially isolated animals share some neural phenotype?

Thank you for pointing out this potential confusion. It is not clear that Buspirone has altered the social preference of anti-social control fish, as we still see ~10% anti-social control fish following drug treatment (Figure 4—figure supplement 1A left panel), similar to the proportion found in an untreated control population. However, given that anti-social fish are rare in normally-raised populations, we do not have enough data to conclude that Buspirone is *only* effective at reducing anxiety in isolated fish.

c) If possible, it would be instructive to perform neural activity imaging on Buspirone treated animals, and in particular, show reduced activity in the optic tectum and PTN, as the authors claim that hyperactivity in these regions is anxiogenic and drives the antisocial behavior of isolates. Thus, if the authors hypothesis is correct, it would be expected that Buspirone treatment, which rescues the behavior, would also reduce activity in these areas.

We fully agree that these are interesting experiments that could shed light on the mechanism of action of Buspirone. We have indeed kept and fixed all the Buspirone treated fish so that we can explore their functional activity by using *c-fosin situs*, however, this will be part of a future study where explore the impact of Buspirone treatment in more detail. Thank you for all the suggestions.

Reviewer #3:This study identifies brains areas that control social behavioural responses towards conspecifics in socially isolated and normally reared zebrafish. I am excited about this work because it opens up the possibility to understand the neurotransmitters and brain areas that control social interactions. The study is short and the data that are presented are clear. In particular, the differences in cfos expression between animals is very impressive.This study would be improved significantly by directly measuring 5-HT levels in these animals (see below). In the absence of this, some of the conclusions are too strong. I also have some suggestions for improving the Discussion section and tidying up the writing.1) This research could be improved by directly measuring 5-HT levels in the hypothalamus of isolated or socially-reared fish. This could be done by HPLC. The caudal hypothalamus contains a number of different neuron types, so IEG activity here could reflect activation of several neurotransmitter systems. In the absence of HPLC data, you need to consider other neurotransmitters that could also contribute to this phenotype.

Thanks for raising this point. We agree that in order to fully understand the impact of social isolation and the mechanism of action of Buspirone, we would need to characterise the levels of 5-HT via HPLC. We are not currently able to do this, but will pursue these experiments for a future publication. Thank you for the suggestion.

We are also planning to measure the expression levels of different hypothalamic neurotransmitters following isolation and after drug treatment. It is already known, indeed, that several of these markers are increased or reduced after isolation, such as Dopamine (Shams et al., 2018) and Serotonin (Shams et al., 2015). These data have been obtained by looking at the level of the expression in different brain areas, but not at single cell resolution. Our behavioural and imaging assay should allow us to achieve this resolution, and we have already started to perform double *in situs* of *c-fos* and candidate markers. These data will also be included in a future publication.

2) The whole brain imaging of cfos is neat, but it loses any cellular resolution. The findings would be strengthened by making sections of the brain and taking higher magnification images. This could include co-labelling with specific markers for dopamine and 5-HT neurons.

We fully agree that it would be very informative to know where and what type of cells are activated during social viewing, after isolation and Buspirone treatment. These data, together with functional imaging, could shed more light on how social isolation affects the social circuit, and how Buspirone is able to rescue the anti-social phenotype. One advantage of using juvenile fish is that their brain is still relatively small, and, therefore, we do not need to do brain sections to achieve cellular resolution. We have indeed started to do double *in situs* for *c-fos* and putative markers, and these data will be included in another publication.

3) The Abstract needs rewriting because the tense used changes throughout.

Thank you for pointing this out, we have corrected the Abstract.

4) The second paragraph of the Introduction (describing isolation and the experimental setup) is not that clear, and can be rewritten to make it easier to follow.

Thank you for your suggestion. Other reviewers also made this point and we have now changed the Introduction to improve clarity and better introduce our main findings.

5) There are a large number of acronyms for brain areas used. It might be better to replace all of these with the full names of the brain areas. This would also improve consistency – for example, you write preoptic (area) but not POA. It will also help to distinguish brain areas from treatment groups (nsc, Pi, Fi) etc.

This is a great suggestion. The text now contains the names of the areas in full, and we have slightly modified the acronyms on the figures to follow the most common conventions found in the literature.

6) Similarly please check name usage throughout – cf preoptic area, preoptic, pre-optic.

Thanks for spotting these inconsistencies. We have changed the text accordingly.

7) I missed the link to sertb expression in the hypothalamus. I would highlight this more clearly, particularly because you use buspirone to rescue behaviour in later experiments.

We agree that this is an important point of the paper, and we have changed the text to emphasise the presence of serotoninergic neurons in the caudal hypothalamus.

8) Some of the discussion points can be improved. There are already studies linking the POA to social behaviour in zebrafish, and these should be included here. It is possible that arginine vasopressin plays an important role in this behaviour. Linking the central hypothalamus to the control of social behaviour is also interesting; the function of sertb positive neurons is well studied. Again, this can be highlighted here.

Thank you for this excellent observation. We have expanded our Discussion and included your main points in the section describing the areas differentially activated in pro- and anti-social fish.

9) It is not clear whether the caudal hypothalamus identified by cfos staining here is the same as the ventromedial hypothalamus described by O'Connell and Hoffmann.

Thanks for highlighting this. Yes, the VMH is the region we see activated in our *c-fos* functional maps. In O’Connell and Hoffmann, 2011, there are two hypothalamic areas already known to be involved in social behaviour: the anterior hypothalamus and the ventromedial hypothalamus (VMH). The VMH is the part of the hypothalamus that is more caudal and comprises the area we see activated in our experiments.

10) Final paragraph of the Discussion I would remove the terms "lonely" and "loner". They are not necessary here and seem anthropomorphic.

We understand that these two terms may not be ideal. However, in accordance with the decision of the Editors, we will keep them, albeit with a more complete explanation of how they apply to our zebrafish experiment.

Reviewer #4:This manuscript describes a series of studies exploring the impacts of social isolation on zebrafish behavior and neural activity. In particular, the authors compare "loner" fish who by choice spend little time socializing with "lonely" fish who are experimentally isolated and discover that while behavior is relatively similar, neural activity is fundamentally different, the latter being rescued by reducing serotonin. In general, I like the paper a lot; it represents a significant effort by the authors and suggests that despite behavioral similarities, all isolation is not the same. This could profoundly shape how we think about sociality in general, and the impacts of isolation in particular. However, there are some aspects of the manuscript that need to be clarified.Substantive concerns:1) One of my big concerns is that the initial paragraph of manuscript does not do a good job of setting up the paper. The first paragraph needs to prepare the reader for the "lonely" vs "loner" fish comparison and explain why it matters. Also, the partial isolation fish are introduced in the second paragraph with no explanation of why they are important. I figured all of this out by the end of the manuscript, but the paper will be much stronger if the reader knows in the first paragraph what to expect. Specifically, I would add conceptual overview that sets the reader up for all of the comparisons and emphasizes why each is needed and important.

Thank you for your great suggestions on how to improve the text. We have now changed the first and second paragraphs of the Introduction to incorporate your comments, please see our additional response to the Editors’ comments.

2) The only places I can find the sample size are in Figure 1 and the paragraph describing the Buspirone treatment. Please provide details on sample size and whether it was sufficiently powered, either throughout or in the Materials and methods section.

Thanks for finding this omission in the paper. We have now added the sample sizes in either the figure panel or legend for all experiments throughout. Our power analysis was based on variance measurements obtained during a previous work investigating the impact of small effect drug treatments on sociality (Dreosti et al., 2015). Given the effect size of Buspirone that we observed, our sample sizes ended up being larger than necessary. This is now explained in the Materials and methods section.